# The Illusion of Competence: A Qualitative Deep Dive into Workplace False Performance

**DOI:** 10.3390/bs14110985

**Published:** 2024-10-23

**Authors:** Marie Frances Dunnion, Gbolahan Gbadamosi, Jan Francis-Smythe

**Affiliations:** 1School of Psychology, University of Birmingham, Edgbaston, Birmingham B15 2TT, UK; 2Department of People and Organisations, Bournemouth University Business School, Bournemouth University, 89 Holdenhurst Road, Bournemouth BH8 8EB, UK; ggbadamosi@bournemouth.ac.uk; 3Worcester Business School, University of Worcester, Castle Street, Worcester WR1 3AS, UK

**Keywords:** counterproductive work behaviour, grounded theory, impression management, job interview, job performance, negative acts, performance appraisal, trust

## Abstract

This qualitative paper develops an understanding of False Performance as a negative form of workplace behaviour which has received scant attention. According to the quantitative literature, which measures False Performance using the Organisational Charlatan Scale (OCS), false performers are incompetent employees who deliberately portray themselves as better able to perform in a job role than they know themselves to be capable. In this study, False Performance was explored in United Kingdom public-sector organisations for the first time, using novel focus-group methodology and grounded theory. Eight focus groups (*n* = 51) were conducted to explore employees’ perceptions and experiences of False Performance, with a focus on job interviews and performance appraisals. Using NVivo for analysis, five categories emerged: Co-Worker Perceptions of False Performance in the Workplace, False Performance in Interview/Appraisal Settings, The Impact of Trust on the False Performer, The Effect of False Performance on Co-Worker Morale, and Preventing False Performance in the Workplace. The results support the literature review that False Performance is a new and negative addition to the taxonomies of job performance. Human resources (HR) managers are, therefore, urged to prevent and reduce False Performance via well-designed selection processes and performance appraisals, thereby improving co-worker trust amongst employees.

## 1. Introduction

This paper takes a novel qualitative approach to exploring the construct of False Performance, which was first introduced when Parnell and Singer [1] developed the Organisational Charlatan Scale (OCS) to address the question, “How does an enterprise identify organisational charlatans?” (p. 443). They coined the term “organisational charlatan” to define those “individuals who seek to improve their perceived performance at the expense of their actual performance” ([1], p. 441). Fox [2] formerly described charlatanism as “the pretension of some practitioners to more knowledge and skill than they possess” (p. 777). Further False Performance research [3,4] has used the synonymous term of “false performer” to refer to the “organisational charlatan”. In this qualitative investigation of False Performance in United Kingdom (UK) public-sector organisations, the newer terminology is adopted. Crucially, this type of incompetent employee deliberately portrays themselves as being better able to perform in a job role than they know themselves to be capable, often engaging in subterfuge (e.g., task avoidance), which can potentially cause harm to both the organisation and the wellbeing of the false performer’s co-workers.

Parnell and Singer [1] explain that the OCS can support human resource management (HRM) decision-making by providing managers with a tool for assessing employees’ tendencies towards False Performance behaviour, separating those who are performance-driven from those who are not. They propose that the false performer functions at a level of incompetence by adopting a range of promotional behaviours to avoid detection and ensure they are not viewed as perfunctory, especially by management. This means that the false performer will likely progress to higher levels of responsibility despite their incompetence, and, in so doing, they will most likely create dissension amongst their co-workers. Parnell and Singer [1] warn managers that they must “learn to identify ‘organisational charlatans’…and remove them from the organisation while preventing other charlatans from joining it” (p. 441). The current study focuses on Parnell and Singer’s [1] warning to HRM and aims to further develop an understanding of False Performance as a construct.

This study represents the first investigation of False Performance in the UK, where it has not previously been examined, thus demonstrating another novel contribution. By employing focus-group methodology, the aim was to uncover the underlying themes and motivations that drive individuals to present a facade of competence or success. The reason public sector organisations were selected was to optimise access to False Performance experiences, given that a combined review of the False Performance and trust literature indicates that there might be a greater incidence of False Performance in the public sector. For example, Gbadamosi et al.’s [4] research suggests that an employee may be more likely to false perform if they have low trust in their manager; and trust research has found that there are likely to be lower levels of trust in public sector versus private sector leaders [5]. Therefore, it tentatively follows that the lower trust-in-management conditions in the public sector may give rise to a greater number of false performers. However, it was not theorised that False Performance in the public sector would be fundamentally different from False Performance in the private sector, taking into consideration that Parnell and Singer [1] designed the OCS as a generic measure of False Performance. Moreover, subsequent research has investigated False Performance in both the public and private sectors, without making a distinction between how False Performance might operate in each sector [3,4].

In their reimagining of performance measurement and management in the public sector, Lewandowski [6] points out that it is crucial to identify failures to ensure future improvements. This encompasses the dark side of the co-production of public services (based on the experiences and knowledge of the service user), a phenomenon known as “co-contamination” [7], which can, in turn, lead to co-destruction [8]. Lewandowski [6] recommends that future research explore how such co-destruction could be linked to the activity of organisational charlatans [1] and how the disclosure of negative performance to the public, including organisational charlatanism/False Performance, could impact negatively on trust. Whilst the current study does not explore the relationship between co-destruction and False Performance, it does focus on False Performance as a negative form of workplace behaviour in the public sector, and it seeks to explore the relationship between False Performance and trust. Critically, Lewandowski’s [6] more recent work demonstrates that there is a continuing appetite for further exploration of the OCS in relation to negative behaviours in the workplace.

Parnell and Singer [1] designed the OCS as a preliminary questionnaire for measuring False Performance, initially administering it to management employees in the southeastern United States. It consists of nine items, e.g., “It’s more important to look busy than to be busy” and “I try to dress better when I’m going to be seen by key organisational decision makers.” Whilst Parnell and Singer [1] obtained alphas of 0.85 and 0.81 in support of the reliability of the OCS, in subsequent studies, Gbadamosi [3] only found an alpha of 0.62, and Gbadamosi et al. [4] reported an alpha of 0.65. As alpha values should be over 0.70 for a scale to be regarded as reliable [9], these findings suggest that future research in this area may require a more robust measure of False Performance. The current paper contends that our qualitative research acts as a valuable precursor to informing any such quantitative revision of the OCS.

While initial questionnaire approaches to the study of False Performance have been valuable, Gbadamosi [3] previously suggested that future research would “benefit from a departure from the traditional survey method to obtain information about how, why and when individuals manifest charlatan behaviour tendencies” (p. 30). Furthermore, Gbadamosi [3] proposed that focus groups might be very useful in this respect, as “it would be valuable to compare such qualitative studies to the little we know from quantitative efforts” (p. 30). Significantly, no other research to date has adopted a qualitative approach to False Performance, so the current study is the first of its kind to directly question work-situated respondents about their perceptions and experiences of False Performance and contribute rich data grounded in real-life employee experiences. Five research questions were examined in this qualitative focus-group study using a semi-structured questioning route:

Primarily, this research study aims to further an understanding of the construct of False Performance, as well as explore the relationship between False Performance behaviour and job performance. Whilst previous research has developed a quantitative measure of False Performance, Parnell and Singer [1] explain that to generate items for the OCS, “an exhaustive set of 92 items believed to reflect attitudinal dimensions of organisational charlatanism were proposed by the researchers” (p. 445). As such, the items which make up the OCS reflect only the researchers’ subjective perceptions of organisational charlatanism. In contrast, the current study will generate rich qualitative data based on the input of management and non-management employees drawn from a variety of public sector organisations—this can be compared to the existing items on the OCS and used for potential future revision of this tool.Secondly, at the conclusion of their study, Parnell and Singer [1] concluded, “In sum the present study has demonstrated that charlatan behaviour is measurable and is associated with performance appraisal. Future research should seek to clarify this critical nexus…” (p. 452). Therefore, the current study seeks to clarify how False Performance operates in the job interview and the performance-appraisal review, something which previous False Performance research has not done.Thirdly, as trust has been shown to have a relationship to False Performance [4], the focus-group study further explores how trust in the workplace potentially affects False Performance.As previous quantitative studies have not explored employees’ subjective experiences of False Performance in the workplace, focus-group participants are asked how they have been personally affected by False Performance in the workplace and whether False Performance has impacted on their work relationships.Finally, inspired by Parnell and Singer [1] posing the question, “What can be done to reduce organisational charlatanism in organizations?” (p. 452), participants are asked for their opinion on preventing/reducing False Performance, encouraging the co-creation of solutions by multiple stakeholders.

## 2. Literature Review

### 2.1. Human Resource Management

This study offers a significant contribution to the field of HRM by highlighting the detrimental impact of False Performance on organisational outcomes. Given the established relationship between HRM practices and individual and organisational performance [10,11], understanding and addressing False Performance is crucial for human resources (HR) professionals. As Guest [10] summarises, organisations that adopt a comprehensive suite of high-performance HRM practices tend to exhibit superior performance metrics, including productivity, labour turnover, and financial indicators. Huselid [11] further elucidates the mechanisms through which HRM practices influence employee performance, emphasising their role in enhancing skills, motivation, and job satisfaction. Moreover, employee performance is a critical driver of organisational success. Given the strong association between HRM practices and performance, this study aims to raise awareness of False Performance as a potentially overlooked and negative aspect of job performance. By understanding the nature and implications of False Performance, HR professionals can develop strategies to detect, manage, and mitigate its detrimental effects.

### 2.2. Job Performance

To support this aim, a study by Ramawickrama et al. [12] will be used to help situate False Performance as a new and negative taxonomy of job performance. Although their research does not specifically refer to False Performance, it does help to explain where False Performance theoretically fits in relation to job performance and its various taxonomies. Ramawickrama et al. [12] define the overall construct of job performance as “the extent to which the employee has shown his or her traits, engaged in behaviours and produced results which are appropriate to task performance, and has engaged in citizenship performance and counterproductive performance during a particular period of time” (p. 77). In terms of taxonomies, they cite Rotundo and Sackett’s [13] three main dimensions of job performance: (1) Task Performance, (2) Citizenship Performance, and (3) Counterproductive Performance. Rotundo and Sackett [13] describe counterproductive performance as a non-task behaviour that has negative consequences for organisations and its employees. In this way, they demonstrate that there is a taxonomy of job performance which accounts for negative work behaviour.

The current study proposes that False Performance is a new addition to the taxonomies of job performance, forming another negative dimension of job performance alongside but distinct from counterproductive performance. It is clear that, whilst both concern unethical work behaviour, they describe entirely different phenomena. Those engaging in counterproductive work behaviours (CWBs) are described as intending to “harm organizations or people in organizations” ([14], p. 151), but this is not the primary motivation of the false performer. The false performer engages in False Performance to conceal their incompetence, and it is only as a side effect of this behaviour that damage may be caused to co-workers and the organisation. Nevertheless, the problems caused by such negative employee behaviour still need to be dealt with by HR managers. For HR managers to better understand what False Performance is, the current study refers to the Impression Management (IM) literature, with the caution that, while impression managers may well be competent performers, false performers are incompetent, skilful only in creating the impression that they perform well.

### 2.3. False Performance Versus Impression Management

IM is defined as the process of establishing favourable perceptions of oneself or one’s ideas in the minds of other individuals [15,16]. Parnell and Singer [1] explain that previous research has generally considered the use of IM techniques without any regard for associated job performance. Therefore, it has not been possible to know whether those obtaining a high IM score are substituting IM for strong performance, nor whether IM is correlated with strong or weak job performance. The fundamental distinction between IM and False Performance is that False Performance comes at the expense of performance, whilst IM does not. IM studies indicate that self-promotion techniques are very likely to feature in a job interview. Barrick et al. [17] explain that IM strategies are used “…to purposefully and strategically present positive information about the self (candidate) in order to obtain a favourable evaluation from the interviewer” (p. 1396). Ellis et al. [18] also report that IM has a significant effect on performance-appraisal ratings. Similarly, Parnell and Singer [1] found significant correlations between management evaluations and OCS scores, suggesting that False Performance may have a positive effect on subjective performance evaluations made in the job interview or performance-appraisal review.

Parnell and Singer [1] clearly distinguish False Performance as a specific type of IM, whilst delineating two types of IM which create a positive effect on performance ratings. The first type of IM is where employees use influence strategies to ingratiate themselves with their managers or other influential figures [19,20]. According to the IM literature, employees who effectively control their image in this positively influential way are more likely to be recruited, be promoted, and receive desirable job assignments than those who do not [21]. The second type of IM relates to how the process of liking or similarity may affect work outcomes [16,22], such that feelings of “like” and “dislike” have a powerful effect on performance evaluations by fostering a “halo effect” [23]. The “halo effect” emerges when a manager forms an initial positive impression of an employee which they then later translate into performance categories, giving rise to future biased evaluations based on initial impressions [24].

Parnell and Singer [1] underline how the empirical research has overlooked the critical issue of performance by focusing primarily on cataloguing IM techniques and their effects on the target audience, e.g., management. To address this gap, Parnell and Singer [1] focused their research on False Performance, thereby creating a connection between IM techniques and job performance. Based on their research and description of the relationship between False Performance and IM, the current study treats False Performance as related to IM but conceptually distinct. Whilst the impression manager and the false performer may engage in similar IM tactics, the false performer’s actions are motivated by their desire to intentionally disguise their incompetence, whereas the impression manager’s behaviours are motivated by their desire to showcase the best of their true abilities.

## 3. Definitional Issues

Whilst the associated literature, especially that on IM, helps to inform an understanding of False Performance, there are some further conceptual boundaries which need to be drawn between IM and False Performance. IM falls into one of two categories: (1) conscious (intentional) or (2) unconscious (unintentional). Levashina and Campion [25] distinguish between IM as the intentional distortion of responses to create a favourable impression, as opposed to self-deception or unintentional distortion of responses. For the purposes of the present research, the conscious process of IM has been juxtaposed against the conscious process of False Performance. False Performance falls into one of two categories: (1) conscious False Performance or (2) unconscious False Performance. The model in Table 1 shows how self-presentation variously combines with competency levels to produce either IM or False Performance. There are four possible scenarios which could occur. Individuals high in self-presentation but low in competence are defined as false performers.

## 4. Unethical Work Behaviour Literature

The construct of False Performance is also related to, but distinct from, prior theories of unethical work behaviour, as is evident from a review of the associated literature. For instance, other researchers have presented immorality in the workplace in terms of the “good soldier” versus the “good actor” [26,27], political behaviours [28], and social loafing [29].

## 5. The False Performer as a “Good Actor”

Analogous to the current study, there is a body of research [26,27], which considers the dichotomy of “positive” and “negative” behaviour in the workplace by outlining the difference between the positively (i.e., genuinely) acting “good soldier” and the negatively (i.e., disingenuously) acting “good actor”. This juxtaposition is helpful for illustrating how “the impression manager” is conceptually distinct from the “false performer”. In the “good soldier”/”good actor” literature, Bolino [26] distinguishes between “good soldiers”, who selflessly engage in organisational citizenship behaviour on behalf of their organisation, versus “good actors”, whose behaviours may be self-serving. In a qualitative interview study of this phenomenon, Snell and Wong [27] asked participants to describe stories about co-workers who were either “good soldiers” or “good actors”. They found that, when distinguishing “good actors” from “good soldiers”, there were two criteria for attribution: wilful behavioural inconsistency, i.e., low generality of behaviour across contexts; and alleged false pretence, i.e., discrepancy between claims and actual deeds. Through comparing 89 “good soldier” stories and 53 “good actor” stories, Snell and Wong [27] found that most “good actor” stories featured alleged wilful behavioural inconsistency or alleged false pretence, or both. In contrast, none of the “good soldier” stories referred to wilful behavioural inconsistency or false pretence. To relate these findings to the present investigation, the false performer can also be seen as a “good actor” who serves their own interests by disguising their incompetence in order to progress in the organisation. There is false pretence involved in their actions because a discrepancy exists between the positive image which they project to others and the actual incompetent job performance which they bring to bear in the workplace.

## 6. Can False Performance Ever Be Positive?

It is possible that False Performance could be incorrectly regarded by some as comprising “positive” political behaviours, which, potentially, allow the false performer to thrive in the workplace and climb the career ladder. Given this potential misinterpretation of False Performance, it is critical to differentiate negative False Performance behaviours from the more positive perspective of “political skill” [30]. Mintzberg [31] first introduced the term “political skill” to describe the social skills which employees may employ in order to survive in an organisational environment. Individuals high in political skill are likely to have the competence to strategically adapt their behaviour to, for example, gain colleague support, assert themselves in negotiations or sales, or acquire access to coveted resources such as technical equipment or budgets [28]. As such, political skill has been defined as “a comprehensive pattern of social competencies, with cognitive, affective, and behavioral manifestations” [32].

To measure political skill behaviours, Ferris et al. [33] designed an 18-item Political Skill Inventory (PSI). The PSI measures four constructs: (1) Interpersonal Influence, (2) Network Building, (3) Social Astuteness, and (4) Genuineness and Sincerity. Some example items include the following: “I understand people well”, “I am good at getting others to respond positively to me”, and “It is easy for me to develop good rapport with most people.” Notably, the four constructs measured by the PSI incorporate positive and honest work behaviours. Thus, the PSI demonstrates that, in certain work situations, employees can use positive political skill to their advantage without engaging in deceptive behaviours. This type of positive political skill can be clearly differentiated from False Performance behaviour, which always involves deception of others to disguise incompetence and avoid detection. False Performance is defined as a negative behaviour, and, unlike an employee practicing positive political skill, the false performer’s motivation for their unethical conduct will always be to mask their incompetence—a purely negative motivating factor which is not implicated in the political skill-literature.

## 7. Social Loafing

Finally, the literature on social loafing has been selected as appropriate for further distinguishing the construct of False Performance because there is a possibility that the false performer’s unethical behaviour could mistakenly be interpreted as a form of loafing. Klotz and Buckley [34] define the phenomenon of social loafing as an employee: “Exerting less effort in the context of a group or team than when working alone” (p. 117). If the false performer’s co-workers or managers recognise that the false performer is underperforming in their job role, they may erroneously attribute the cause to loafing rather than incompetence and deliberate deception. Crucially, Klotz and Buckley [34] explain, “It is important to note that social loafing differs from other forms of counterproductive work behaviour (CWB-O) in that employees engaged in loafing do not consciously and deliberately withhold effort from a performance situation. It is a naturally occurring phenomenon. Instead, the mere presence of others doing the same task can be enough to facilitate social loafing” (p. 124).

So, whilst employees engaged in loafing are not deliberately withholding effort, in contrast, the false performer is consciously and purposely withholding effort because they lack the competence to carry out the job tasks which they claim to be able to accomplish. Therefore, there is a clear distinction between false performers, who lack competence, versus social loafers, who are merely neglecting to apply their competence. Furthermore, social loafing is specific to employees withholding effort only when they are working as part of a group. On the other hand, the false performer will consistently perform below par, whether working individually or as part of a team. This is because the false performer does not have the competence to better their performance when working alone, whereas the loafing employee does. Overall, employees who loaf in groups do so because they unconsciously allow themselves to be carried along by the effort of the collective, but the false performer underperforms in a group because they are incompetent and consciously aware that their co-workers’ efforts will help them to disguise their incompetence. This is a crucial distinction that management should be aware of when trying to identify False Performance so that they can adapt their management of each behaviour accordingly.

## 8. False Performance and Management

### 8.1. Mushroom-Type Management

Recent research by Ergün [35] has examined the effect of mushroom-type management behaviour on teachers’ organisational loneliness and organisational charlatan/False Performance behaviours. Mushroom-type managers have been defined by Tekin and Birincioğlu [36] as managers who leave their employees in the dark (like mushrooms), giving them only instructions rather than full background information. It is possible that such mushroom-type management behaviour leads to teachers experiencing loneliness in the school environment. Therefore, Ergün [35] posed the following research questions: (1) Does mushroom-type management behaviour predict teachers’ organisational loneliness? and (2) Does mushroom-type management behaviour predict teachers’ organisational charlatan behaviours? According to findings, mushroom-type management behaviour significantly predicted organisational charlatan behaviour at a positive low level; it also significantly predicted organisational loneliness behaviour at a positive low level. It appears that by management not sharing information and involving employees in decision-making, this may positively encourage False Performance. For example, employees may respond to their manager’s superficial instructions by portraying themselves as more competent than they are so that their performance is perceived to be better than that of others. In this way, it is possible that a false performing teacher may attract positive evaluations and rewards despite their incompetent performance on the job. In this climate of inauthentic communication and performance, teachers may feel more alone, and there is the risk that False Performance and employee loneliness will negatively impact on organisational goals. In conclusion, Ergün [35] recommends that mushroom-type management behaviours be avoided or used sparingly to prevent teachers being pushed towards False Performance and/or loneliness; instead, managers should adopt a more participatory and transparent approach.

### 8.2. Trust and Shame

Since Parnell and Singer’s [1] research, two key studies [3,4] have provided further evidence of construct validity for the OCS, as well as examined several associated variables, many linked to management. At the outset of their study, Gbadamosi et al. [4] asked, “Could the absence of trust in management among employees increase the incidence of charlatan behaviour among them?” (p. 754). To explore this research question, Gbadamosi et al. [4] examined the relationship between False Performance and trust in management. They also examined the relationship between continuance commitment (people staying with the organisation because they need to) and False Performance. Continuance commitment is when an employee remains with an organisation because they cannot find equal employment elsewhere or they perceive high costs of leaving. Meyer and Allen [37] have distinguished this need to stay with the organisation (continuance commitment) from a desire (affective commitment) and obligation (normative commitment) to maintain employment. In Gbadamosi et al.’s [4] study, both trust in management and continuance commitment emerged as significant predictors of False Performance (whereas affective and normative commitments were not found to correlate significantly with False Performance). In summary, they found an inverse and significant relationship between False Performance and trust in management, and a positive and significant relationship between False Performance and continuance commitment. Based on these results, Gbadamosi et al. [4] concluded that the lower the trust the employee has in the organisation, the lower their likely commitment and the greater the likelihood they will, therefore, engage in False Performance.

Similarly, in the other direction, recent research suggests that the higher the trust an employee has in the organisation, the better their job performance is likely to be. Ayça [38] underscores the significant impact of authentic leadership, trust in the supervisor, and trust in the organisation on employee job performance. The key implication of Ayça’s [38] study is the importance of authentic leadership, which prioritises fairness in managerial methods, actions, and activities, and its significant role in enhancing job performance. Job performance in organisations can be understood as the unique contribution each employee makes through their work, which is crucial for the organisation’s overall success [38]. The perception of trust between employees and the manager within the organisation acts as a cohesive force, akin to a robust corporate culture. As such, trust in an organisation and its managers plays a pivotal role in boosting employee motivation and fostering positive relations between employees and managers, which can significantly impact overall job performance and success of the organisation [38]. Given Ayça’s [38] up-to-date research, coupled with Gbadamosi et al.’s [4] findings, trust is strongly implicated as having a key role to play in positively or negatively affecting job performance via the work relationships which develop between co-workers.

Another variable which has been identified as playing a potential role in positively influencing trust, co-worker relationships, and job performance is social media [39]. Kasim et al. [39] investigated the role of social media use at work and found that its usage significantly predicts social capital (online social network ties and trust), which then positively promotes work engagement and innovative job performance. Research suggests that social media has become a platform for fostering informal relationships among organisational members, thereby encouraging the development of trust in the workplace [40]. Prior studies have also shown that social media use not only significantly contributes to developing workplace trust among employees but also offers various benefits for organisational behaviour, highlighting its capacity for promoting positive change and a sense of optimism [41,42,43]. Whilst the current study did not directly examine the relationship between social media and trust/co-worker relationships, the grounded-theory methods used meant that there was scope for social media to emerge for exploration as a moderator between trust and job performance.

Similarly, although the current study was conducted before the COVID-19 pandemic, there is an underlying awareness of the vastly changed working landscape and the influence of remote or hybrid work practices on employees’ job performance. For example, Kifor et al. [44] investigated the impact of remote work on employees’ self-assessed performance when working from home during the COVID-19 pandemic. Their findings show that trust in management generated a negative effect on employees’ social performance (e.g., getting along with others at work and avoiding arguing with others), meaning that those with weak trust in management were found to rely more on their co-workers, thus performing efficiently as a team to get the job done without management input. Conversely, trust in management was not found to be associated with technical performance (e.g., handling the responsibilities and daily demands of work, performing work-related duties without mistakes, and fulfilling the performance criteria demanded of a job). This latter finding is consistent with previous studies which have reported that trust in management generates positive or no effect on job performance [45,46]. It is also reminiscent of Gbadamosi’s [3] earlier study in which False Performance was not found to be significantly corrected with trust in management. This is, of course, contradictory to Gbadamosi et al.’s [4] later finding of an inverse and significant relationship between False Performance and trust in management. The current study was, therefore, keen to re-examine trust to clarify its relationship to False Performance.

Moreover, the current study aimed to explore for the first time whether co-worker trust negatively or positively affects False Performance. Parnell and Singer [1] point out that, since false performers do not appear to fool their co-workers easily, increased attention may need to be directed at using the co-worker to identify the false performer. Gbadamosi [3] also identified co-worker trust as a concept which may be able to further an understanding of the variables related to False Performance. For the purposes of the current research, trust in co-workers is defined as the willingness of an employee to be vulnerable to the actions of fellow workers, whose behaviour is not under their control [47]. For example, is the false performer *less* likely to indulge in False Performance the more that they trust (are vulnerable to) their colleagues? Or, conversely, will they be *more* likely to engage in False Performance if they believe that their co-workers are trustworthy? Yakovleva et al. [48] explain that low-trusting individuals frequently become exploitative in instances where they experience their opponents behaving cooperatively over a long period. The current study aims to discover whether co-worker trust is likely to negatively affect the false performer’s behaviour in this way or more likely to have a positive effect on the false performer’s actions. In some studies, trust in co-workers has been found to be positively related to job performance [47,49].

Additionally, the current study will consider whether the relationship between False Performance and trust could be moderated by the false performer’s feelings around their own unethical behaviour. For example, is it possible for the false performer to feel some degree of shame about their deceit which could then lead them to reduce their False Performance behaviours (perhaps leading to improved co-worker trust)? With a focus on corruption in developing countries, Abraham and Berline [50] explored organisational charlatan/False Performance behaviour as a predictor of shame proneness, with shame defined as negative feeling as a result of moral breach. Whilst their study found that False Performance was unable to predict shame-negative self-evaluation (shame proneness), their hypothesis and results encourage contemplation of whether the false performer is likely to feel shame around their disingenuous actions or whether they engage in False Performance because they feel little shame. The current study may be able to provide further insights into the false performer’s feelings, including those of shame, and how this might affect their False Performance and relate to different types of trust in the workplace.

## 9. Method

This study adopted a qualitative focus-group method (e.g., [51]) to encourage both management and non-management participants to speak about their perceptions and lived experiences of False Performance without imposing preconceived notions upon the group. Focus-group methodology was also adopted because of the compelling evidence to suggest that “focus groups reach the parts that other methods cannot reach—revealing dimensions of understanding that often remain untapped by the more conventional one-to-one interview or questionnaire” ([52], p. 109).

Focus groups are commonly used in qualitative research, and Krueger and Casey [53] explain that a focus-group study is a planned series of discussions designed to obtain perceptions on a specific area of interest in a permissive, non-threatening environment. According to Morgan [54], “The hallmark of focus groups is the explicit use of group interaction to produce data and insights that would be less accessible without the interaction found in a group” (p. 12). Moreover, Kitzinger [52] points out that group work is invaluable in the development of grounded theory because, in line with Glaser and Strauss’s [55] initial conception, it focuses on the generation rather than the testing of theory in its exploration of participants’ experience. Also, the key to selecting focus-group methodology for the current study was its appropriateness for researching the sensitive organisational issue of False Performance. Morgan and Krueger [56] emphasise the unique ability of focus groups to convey a humane sensitivity, a willingness to listen without defensiveness, and a respect for opposing views. Krueger and Casey [53], likewise, describe how focus groups are designed to promote self-disclosure by creating a relaxed environment in which participants feel free to express their opinions without being judged. Therefore, the focus group was deemed to be the ideal space for the flexible, but systematic, collection of grounded-theory data on the topic of False Performance.

### 9.1. Participants

A total of 51 employees were recruited from four UK local government organisations in the public sector, and eight focus groups were then held in these organisations (four management, *n* = 26; four non-management, *n* = 25). Of the participants, 41.2% were male and 58.8% were female. Table 2 provides demographic information for each of the focus groups. The only prerequisite for participation was that each employee had at least two years’ work experience to ensure a sufficient work history to make meaningful contributions. Table 2 shows that both management and non-management had, on average, over 20 years’ work experience on which to base their discussion contributions.

### 9.2. Data Collection

All focus groups were approximately 90 min in duration and limited to nine participants (management-group sizes, seven, eight, nine, and two; non-management seven, eight, seven, and three). Focus groups were divided by status to create an environment in which participants would feel comfortable expressing their opinions with no power differentials [53]. At the start of each session, participants were assured of confidentiality and introduced to the purpose and ground rules of the discussion. Using the guidelines of Krueger and Casey [53], a pre-determined question format was followed, and participants were first asked, “What is your understanding and experience of people who practice False Performance in the workplace?” Additional questions further explored participants’ experiences and opinions of False Performance in the job or performance-appraisal review. After this, participants were asked to produce a written list of False Performance behaviours. Remaining questions asked participants how False Performance affects work relationships and how trust in management or co-worker trust influences False Performance. Participants were also asked if they could provide a solution to the problem of False Performance.

### 9.3. Data Analysis

Focus-group discussions were transcribed from the audio recordings and content analysed using grounded theory [55] because it has the flexibility to allow new themes to emerge for exploration. Goulding [57] suggests its use when the topic of interest has been relatively ignored in the literature, as has the construct of False Performance. Charmaz [58] summarises, “Stated simply, grounded theory methods consist of systematic, yet flexible guidelines for collecting and analysing qualitative data to construct theories ‘grounded’ in the data themselves” (p. 2). Preconceived ideas are not forced upon the data, as the data should be allowed to speak to the researcher. Glaser and Strauss [55] explain, “grounded theory is derived from data and then illustrated by characteristic examples of data” (p. 5). Accordingly, the current study used data to illustrate the research findings.

The first stage of grounded-theory analysis usually consists of initial and focused coding, and, for many grounded theorists, line-by-line coding is the first step [58]. This requires naming every line of the written data [59]. A less time-consuming approach, suggested by Glaser [60], involves transcribing one interview and then listening to tapes to identify codes and themes. In the present study, all interviews were fully transcribed, and, after conducting a line-by-line analysis with the first two transcripts, the process of focused coding was entered into for the remaining six transcripts. The qualitative data-analysis software NVivo 9 [61] supported the coding and comparison of textual passages.

Focused codes are more directed, selective, and conceptual than line-by-line coding [59]. During focused coding in the current study, the most useful initial codes were tested against the extensive data to identify which ones made the most analytic sense for categorising the data incisively and completely. This resulted in a series of refined codes which were developed with the use of memos. As part of focused coding, themes were categorised into a broad category or subcategory. The current study employed Charmaz’s [58] procedure for axial coding, developing the subcategories of a category by showing the links between them as more was learnt about the experiences the categories represented.

## 10. Results and Interpretations

During the process of data analysis, it became apparent that False Performance is perceived as an important issue within organisations. Across all eight focus groups, there were data to suggest that False Performance is a behaviour which participants observe regularly, is something that resonates with them, and is an issue to which more attention should be paid. For example, the following non-management focus-group participant highlighted the frequency of False Performance experienced in their organisation:

And I’ve seen a lot of cases, I mean I’ve been in [the organisation] for twenty-two years, I’ve seen a lot of cases where the term “promoted beyond your own competence levels” happens a lot.
**[Non-management, FG06]**


A management participant similarly explained that a lot of false performers had been admitted into their organisation as a result of candidates performing really well in the job interview but then being unable to perform in the way described when actually in the job:

Sound like they can do a good job, they can do this and do that, until they’re actually in the job. We find that a lot, don’t we? They can’t actually, when they’re physically doing it, they can’t do it.
**[Management, FG01]**


During analysis, data were coded according to whether participants were of management or non-management status. Five primary categories emerged across both groups. These were Co-Worker Perceptions of False Performance in the Workplace, False Performance in Interview/Appraisal Settings, The Impact of Trust on the False Performer, The Effect of False Performance on Co-Worker Morale, and Preventing False Performance in the Workplace. Various subcategories were then related to these main categories. A summary of the major categories, subcategories and focus-group narratives on False Performance, can be found in Table 3.

## 11. Co-Worker Perceptions of False Performance in the Workplace

The majority of focus-group participants enthusiastically discussed their perceptions of False Performance in the workplace through various means, such as storytelling. Participants generally perceived False Performance as being about concealment. One manager explained as follows:

I tend to think of somebody who’s got a facade, who’s hiding things, and the temptation is always to try and find out what they’re hiding. Not really listen to what they’re talking about or doing, the action. It’s about, for me, it’s about what’s the hidden bit?
**[Management FG04]**


The concept central to False Performance is the idea of there being a disparity, i.e., “a hidden bit” between the false performer’s self-presentation and their actual job performance. In the focus groups, there was the shared idea that false performers deliberately use one set of words or actions for the purpose of creating a positive impression which differs from their actual job performance. Participants provided examples of behaviours which they considered specific to false performers, such as “backstabbing” (betraying a co-worker), pretending to look busy, telling tales to sabotage someone else for personal gain, and the boss who over-delegates work to gain favour with their own superiors. One non-manager pointed out that the false performer could be the boss in the following exchange:

**Participant 1:** You report them (the false performer) to the boss, and they don’t do nothing about it.

**Participant 2:** But they could be one, you know, one of the bosses themselves.


**[Non-management, FG02]**


In this category, the main subcategories emerged as Claiming Credit for Others’ Work, Boss Over-Delegation to Subordinates, and Shifting the Blame.

### 11.1. Claiming Credit for Others’ Work

This was discussed in terms of the false performer taking credit for another employee’s work, as well as that of the entire team. For example, one manager gave an insight into this False Performance behaviour by sharing the following story:

It’s when you see somebody else’s signature on the bottom of something…Certainly, over the years, within the local authority, you’re asked to write something, reports mainly, and it involved a lot of work and then someone else’s signature goes on the bottom of it. And you think, come on, there’s not even a reference to yourself.
**[Management, FG04]**


Another manager also disclosed how they had previously done work for a false performer who then claimed full credit for the results. They explained, “…because obviously the work that I was doing for her, her boss didn’t even know that half of the work she was presenting wasn’t even done by her” **[Management, FG01]**.

It is in such ways that the false performer claims credit for their colleagues’ work to appear competent to management.

### 11.2. Boss Over-Delegation to Subordinates

This issue emerged across all non-management groups as one of the most prevalent examples of False Performance, with one non-manager explaining it as follows:

I think it depends how performance is perceived as well. Because the manager might look like they’re performing well because they’ve been taking the praise and saying that they’ve been doing the work, so they might look like they’re successful when actually it’s not them that have done the work.
**[Non-management, FG03]**


Another participant in the same focus group agreed in direct response by saying, “It’s the people that they’ve delegated it to” **[non-management, FG03]**. Although managers have to assign responsibilities to their staff, reasonable delegation becomes False Performance when the manager over-delegates work in order to mask their own incompetence. One manager shared the following example scenario:

The boss is…taking all the credit and sit backs…maybe hasn’t done the paperwork or anything to go with it or hasn’t put all the file work together and their secretary has, and then they’re saying, “Well, it was all me.”
**[Management, FG01]**


A non-manager spoke about the phenomenon of *Boss Over-Delegation to Subordinates/Claiming Credit for Others’ Work* by sharing a specific example of False Performance behaviour within their organisation. However, they then questioned whether an individual is personally accountable for False Performance or whether it is the failing of an organisational culture which encourages such behaviour:

There are a lot of individuals who sit there and do their [performance appraisal review] and they will claim the credit for a piece of work or a number of pieces of work. I work in the Ops department part-time at the moment, and there should be five staff in there—at the moment there’s only three of us. In there, we will do all the groundwork for a particular project, we’ll present all that groundwork to the line manager. The line manager is the one that will present it to the Senior Management Board. The line manager is the one that will get the credit for the work that’s been carried out. The line manager has not necessarily had anything to do with any of it whatsoever; he’s presented the work that we’ve done, but he will get the credit for that. Now I know for a fact, in this organisation, that a number of people have been promoted on the back of work that’s been done that isn’t necessarily work that they’ve done, it’s just work that they’ve taken the credit for. And that’s…You can say that’s false performance, you could also say it’s good management. At the end of the day, he’s getting what he wants out of his staff and he’s presenting it the way he wants to the way that our Management Board works. And that’s more a failing of the management than that individual.
**[Non-management, FG06]**


Management participants discussed how this False Performance behaviour could hinder an employee’s career prospects, explaining that a subordinate will miss out on recognition for work that their boss is claiming credit for. They even suggested that a manager might sabotage an employee’s chances of promotion to keep them in the subordinate role, where the manager can continue to claim credit for that employee’s work in order to impress their own superiors.

### 11.3. Shifting the Blame

Participants suggested that false performers blame other individuals for their mistakes. As the false performer never takes any responsibility for unsuccessful work outputs, this allows them to maintain a positive appearance of competent conduct whilst others carry the blame. One manager said,

I think this person, again, is somebody who tends to blame others and doesn’t accept ownership of his responsibilities in decision-making and workloads. Will actually say things like, ‘nobody told me’.
**[Management, FG04]**


A manager in a different focus group also suggested that False Performance behaviour involves shifting the blame, explaining that this is to cover wrongdoing: “If they [the false performer] know something’s gone wrong in the workplace, shifting the blame onto somebody else” **[management, FG01]**. There was general agreement from other participants in the group, who replied, “Yeah, yeah, yeah.” And, in another management group, a participant suggested that the false performer will always specifically blame their manager:

Always saying that the manager’s got it wrong or it’s the manager’s fault or somehow the manager hasn’t given them enough information to do the job or something; somehow, it’ll be the manager’s fault.
**[Management, FG08]**


Across the focus groups, there was also some suggestions that the false performer might shift the blame to other factors, like technology or car problems, in order to escape detection. In one management group, someone also proposed the following: “Hiding behind illness and having lots of sick leave” **[management, FG08]**. One non-manager astutely summed up the false performer as being all about “passing the blame but taking the praise” **[non-management, FG03]**. And a non-manager in another focus group summarised by saying, “Shifting blame. Make sure nothing sticks. Teflon” **[non-management, FG06]**.

This subcategory calls for a distinction to be made between the false performer who makes excuses for False Performance versus the employee who makes excuses for poor performance. Sonnentag and Frese [62] explain that job performance does not remain stable over time, with variability dependent on learning processes and other temporary changes in performance. The difference between the false performer and their co-workers is that the latter will learn over time and improve their job performance, whereas the false performer will not.

## 12. False Performance in Interview/Appraisal Settings

Participants discussed the False Performance behaviours which might occur in the job interview, including lying about qualifications, exaggerating, dressing to impress, and presenting a fabricated Curriculum Vitae (CV). They also discussed those which might occur in the performance-appraisal review, such as ingratiating oneself with the reviewer/manager, exaggerating, and name-dropping (mentioning impressive associates). The main subcategories are Lying About Qualifications, Over-Talking as a Smoke Screen, and Claiming Credit for Others’ Work (1:1).

### 12.1. Lying About Qualifications

Interview-based False Performance behaviours were discussed extensively in all groups. One manager commented on how the false performer’s fabricated CV can help them to secure a position:

They exaggerate, they exaggerate, you know. They probably say they’ve got more GCSEs, more A-levels and all that. Probably not. But, because it’s there and it looks good and they’re giving this overall confident interview, they’re going to go, “Oh great, we’ll have him…or her.”
**[Management FG01]**


A non-manager explained why this type of False Performance is likely to be detrimental to the organisation:

Well, if they’ve got a job based on qualifications they’ve lied about, their performance in the job is going to be limited because they’re not going to have the skills to do the job, the skills required.
**[Non-Management, FG03]**


False qualifications are unlikely to reflect the truth of a candidate’s qualifications, skills, or experience. Double-checking qualifications at the interview stage may be one of the easiest ways for HR managers to detect False Performance early on. However, even this may not be entirely foolproof as one non-manager described how, in their experience, they have encountered candidates with the right qualifications but insufficient experience. This clearly underlines how important it is for HR managers to carefully appraise both qualifications and work experience:

So, although they had the qualifications on paper, they were actually lacking in the experience which was a concern that I had at the time. And eventually I think the individual ended up leaving the organisation. But if you’d sat with them in that interview and spoke to them, you’d have thought, ‘Wow, this person is really right for this job.’ But then, once they were actually in the role, it was a completely different story.
**[Non-Management, FG06]**


Moreover, focus-group data suggest that candidates might also lie about their past work experience in the job interview. In one focus group, a manager gave a detailed example as follows:

I know, outright lies is one. Yes, I can think of one case of someone who was forced out of [name of organisation]. I only found this out later, but they were told that they had to leave, but they were allowed to resign and that was what made it particularly difficult, they weren’t sacked…they were told you either leave or we will sack you, but they chose to leave and to resign. They then went to the next job at the [organisation] where I was working and when asked the direct question, ‘Why did you leave that [previous organisation]?’ They said, ‘Oh they were tediously administrative…I was being confined in that particular job.’ And I only later discovered, from getting to know the people at [previous organisation], that it was just a plain, outright lie, that she’d had a terrible experience, that she’d almost financially ruined the department and was extremely unpopular and a minuted Committee Meeting actually agreed, ‘She has to go.’ And she just said none of that in her selection meeting so just plain, outright lies, but lying very, very convincingly; she was a very skilled communicator.’
**[Management, FG08]**


This example indicates that it can be very difficult for interviewers/HR managers to detect False Performance during the job interview, particularly if the false performer misrepresents their past performance and there is no way of fact-checking and/or reason for querying the information the false-performing candidate has provided to the interview panel.

### 12.2. Over-Talking as a Smoke Screen

Both management and non-management participants suggested that the false performer is likely to divert discussion away from their incompetent job performance in the performance-appraisal review by over-talking as a smoke screen. One non-manager explained this behaviour as follows:

I guess in some ways another characteristic they might have is that, at the beginning, they just talk and talk and talk, and say all the good things that they’ve done or give loads of evidence, and not allow for any questions, just because they want to get as much in there as possible. And then, you know…they think, ‘Oh well, there’s the half an hour up.’
**[Non-management, FG07]**


A management conversation similarly explored how the false performer might skilfully create a “wall on words” in their performance appraisal to divert and avoid detection of their incompetence:

**Participant 1:** And they will talk at great length and not allow you to.

**Participant 2:** Yes, that’s certainly common in my experience…They’re [false performers] very good at communication and can impress people by talking skilfully.

**Participant 1:** Yeah, absolutely.

**Participant 2:** And so, they can create a wonderful impression based upon not very much. So, what underlies it may not be very much, but they’re very skilful at making a great deal out of a little…and, you know, skilled communicators. But not having other perhaps more substantial or job-related skills, something like that. So yeah, creating a kind of wall of words, if you like.


**[Management, FG08]**


By magnifying the “positives” of their work, especially what they claim to be past victories, the false performer may succeed in preventing their manager from detecting their current False Performance. If they fill up the performance-appraisal time with talk of their self-proclaimed achievements, this then reduces the opportunity that the interviewer has to present them with any challenging questions about their actual incompetent job performance.

### 12.3. Claiming Credit for Others’ Work (1:1)

This issue was raised by both management and non-management. One non-manager commented as follows:

I think they would take the credit for pieces of work or things that have happened they have not been involved in that somebody else has been. I think that’s quite common that somebody takes the credit for projects or things that have happened, and I’ve seen that happen a lot where people have done a piece of work and then somebody else has taken all the glory for it.
**[Non-management, FG06]**


Another non-manager suggested that the private format of the one-to-one performance-appraisal review makes it easier for the false performer to claim credit for others’ work:

One-to-one, you can much more easily manipulate that, put the message across that ‘I did this’ and ‘I drove this,’ ‘I instigated this,’ and ‘This was all my idea and I drove my team to do it.’ When it may be you didn’t even know anything about it until the results came in at the end and then you say, ‘Oh look, look what I’ve done.’
**[Non-management, FG6]**


Without fear of contradiction from co-workers, the false performer can manipulate the information which they give in the one-to-one performance-appraisal review to make themselves appear more competent.

On the positive side, however, a non-management participant indicated that one-to-one performance appraisal reviews could provide a good opportunity for co-workers to bring the false performer’s behaviours to the attention of management; however, many employees may be too frightened of the repercussions of doing so:

You know when you have your [performance appraisal review abbreviation], you should bring it up then, but you’re frightened to say anything because of what will happen when they [the false performer] finds out…This is what the [performance appraisal review abbreviation] is about…so that you can say how you feel, but you’re frightened to do it. I mean there’s not just you, the others are frightened to do it because it’s going to come back on you.
**[Non-management, FG02]**


This non-management insight reveals that it can be incredibly difficult for HR to gain evidence of the false performer’s behaviour because of co-worker reluctance to report it to management/HR. To counteract this tendency, the performance-appraisal review needs to be designed and facilitated so that employees feel that they can share all of their concerns in a safe space. One simple way to achieve this is to ensure that performance-appraisal reviews are always performed one-to-one, despite the proposed risk of false performers claiming credit for others’ work (1:1); in this example, the non-manager explained that performance appraisals were often done in a group. This would be problematic if a false performer were part of the group because it is highly unlikely that co-workers would feel comfortable to openly report their colleague’s False Performance in this collective space.

## 13. The Impact of Trust on the False Performer

Co-worker trust was discussed in terms of team trust being essential, trust taking time, and the false performer pretending to trust others. Trust in management discussions covered issues such as the false performer manipulating a trusting manager and uncertainty as to whether trust would affect False Performance. Subcategories emerged as Trust is Not a Concept for the False Performer, Co-Worker Trust Breeds False Performance, Clarifying How Trust Relates to False Performance, and Co-Worker Trust Reduces False Performance.

### 13.1. Trust Is Not a Concept for the False Performer

Management and non-management both reasoned that, as the false performer is not familiar with trusting others or being trusted, trust will not be a concept which will have relevance for them. One manager stated the following:

If it is the false performer and they know they’re being deceptive then, if that’s their persona, which is built upon a lie if you like, then you’ll have a level of distrust in everybody around you because you’d think they might be acting in the same way. So, I would think, if an individual acts in those manners, I would think they’d have a very limited amount of trust for anybody.
**[Management, FG05]**


From this perspective, the false performer will not trust their manager or co-workers because they simply do not trust anybody. Due to their own disposition of untrustworthiness, the false performer is liable to think that the world comprises other false performers, so trust will not affect their propensity to false perform either more or less.

A non-manager similarly expressed that trust would be irrelevant to the false performer because, regardless of their felt trust in co-workers, they are focused on opportunities for manipulating others in the workplace into believing their masquerade of competence:

I agree with you because I think the false performer is not…It is irrelevant whether they trust anybody else or not because what they’re looking for are opportunities to exploit and manipulate. So, I don’t think trust comes [into it]…I think they’ll be looking for the next opportunity and the next person that they can actually draw into this charade.
**[Non-management, FG06]**


### 13.2. Co-Worker Trust Breeds False Performance

There was mixed response to co-worker trust. Non-management mostly opined that a trustworthy co-worker would encourage False Performance, whilst only one management participant expressed this viewpoint. When reviewing a list of False Performance behaviours generated by the group, one non-manager reached the conclusion that co-worker trust is likely to encourage more False Performance:

Looking back at the list of the behaviours of false performers, I think they’re more likely to implement those behaviours if they think that their contemporaries are telling the truth.
**[Non-management, FG07]**


Another non-manager similarly proposed that the false performer is likely to take advantage of trustworthy co-workers: “I think they’d walk all over you” **[non-management, FG03]**. If the false performer interprets their colleagues’ trust as an easy way to manipulate them, this subcategory suggests that co-worker trust could actually breed False Performance.

### 13.3. Clarifying How Trust Relates to False Performance

The relationship between trust and False Performance confused both management and non-management. For instance, one non-manager asked, “Is it trust in their colleagues’ ability, or is it trust in…they’re not going to find me out, that I need to keep this deceit up?” **[non-management, FG07].**

This participant was trying to clarify whether co-worker trust is about the false performer trusting their colleagues or about them trusting that their co-workers will not detect their False Performance. With definitions of trust varying across focus groups, it may be that the confusion around trust is attributable to the grounded-theory methods used. To allow the data to reveal themselves, the focus-group moderator did not define trust but rather explored it with more flexible questioning. Alternatively, it could indicate that there is not a meaningful relationship between trust and False Performance, reflective of Gbadamosi’s [3] research, which did not find a significant relationship between the two.

In one non-management focus group, a participant suggested that the relationship between trust and False Performance is probably influenced by environmental factors; for example, the False Performance–trust relationship may be dependent on the culture, country, and type of industry the false performer is working in:

So, it’s probably a cultural thing as well. In different industries, it will be different…And probably, in different countries, it’s more acceptable to lie.
**[Non-management, FG07]**


This suggests an additional and interesting lens through which to view False Performance and its relationship to trust, and it also raises the question of whether False Performance is more permissible and/or encouraged in some cultures and countries and, if so, whether this is on a subtle or overt level.

### 13.4. Co-Worker Trust Reduces False Performance

This subcategory contradicts that of Co-Worker Trust Breeds False Performance by suggesting that, if the false performer perceives themselves to be trusted by co-workers, this might encourage them to reduce their False Performance behaviours. The false performer may feel guilty about behaving deceptively when those around them are acting in trustworthy ways. A non-manager spoke of the effect in this way: “So if they had full trust in someone, they might not want to stamp on that person ’cause they might like them, do you know what I mean?” **[non-management FG03]**. This participant reasoned that co-worker trust will help the false performer to build up relationships, and this, in turn, will prevent them from manipulating colleagues for their own gain.

As two opposing subcategories indicate that co-worker trust could breed versus reduce False Performance, possible moderators of this relationship could be (1) feelings of shame and/or guilt and (2) feelings of liking for co-workers. Assuming a positive relationship between co-worker trust and False Performance, co-worker trust will promote an increase in False Performance (Co-worker Trust Breeds False Performance). However, this relationship could be moderated by the false performer’s feelings of shame/guilt or liking for co-workers, and, in this situation, co-worker trust will lead to a decrease in False Performance; thus, the relationship between co-worker trust and False Performance becomes negative (Co-worker Trust Reduces False Performance). However, future quantitative research would be necessary to examine these moderating variables.

Alternatively, one manager suggested that both scenarios may play out, or at least appear to, with the false performer engaging in reciprocal trusting work relationships for a period, but these then breaking down:

The charlatan seems to operate in cycles. You know, there will be lots of trust and they’ll be saying the right things and noises, and suddenly you find you’re being stabbed in the back and then…they’ve got over their problems, and they’re back on side, and actually it [False Performance] goes down again. And you soon identify that person, you soon know who that person is, and whether you can trust them or not and, for a while, you may think you can trust them but it’s like…on what terms?
**[Management, FG04]**


So, even in some situations where co-worker trust seems to lead to a reduction in False Performance, there is a question mark over whether the false performer is genuinely adapting their behaviour due to, for example, liking for co-workers, or whether they are just “saying the right things” to give the impression of mutual trust.

## 14. The Effect of False Performance on Co-Worker Morale

Participants spoke about the demoralising effect of False Performance on the team, questioning why they should do the job when the false performer does not have to. Other issues covered were the disruptive influence of the false performer and the strain, resentment, and lack of trust created by the false performer’s actions. The two subcategories emerged as Reluctance to Report False Performance and Bad for Morale.

### 14.1. Reluctance to Report False Performance

This was primarily discussed by non-management, as they are usually the ones who have to report False Performance to management, as depicted here:

Yeah. Well, even when I did, I got branded as the black sheep (i.e., odd one out) of the group because I challenged this particular person. And it’s sometimes just best to keep quiet, keep your head down and just get on with it.
**[Non-management FG03]**


If employees believe that by challenging a false performer, they themselves will be disadvantaged in the process, then it is unlikely that they will report False Performance to the HR department. The likely outcome of this is that employees will endure the false performer’s misconduct in silence. As one manager explained, “…it takes a lot of strength and courage in an individual to try and do something about someone they perceive to be the unethical charlatan” **[Management FG04]**.

### 14.2. Bad for Morale

Non-management spoke most extensively about the demoralising effects of False Performance, with one explaining how feelings of resentment might arise in those who detect False Performance in their co-workers:

I think it’s demoralising to others that know that they’re false performing, especially if they see them doing well, and if they’re getting promoted…that can be really demoralising for the others, especially contemporaries at a similar level. If a false performer seems to be doing well, it can either make you want to false perform. Well, they’re false performing and getting no reprimanding. In fact, they’re doing well and better than me because I’m being honest. You’re more likely to false perform.
**[Non-management, FG07]**


This participant expressed concern that False Performance might lead to a contagion effect within the team. If co-workers observe the false performer earning promotion, they may re-evaluate their own honest, competent conduct and instead adopt False Performance as a means of climbing the career ladder. Even if co-workers choose not to false perform, this subcategory and that of Reluctance to Report False Performance indicate how False Performance could detrimentally affect feelings of co-worker trust and trust in management to deal with False Performance. It is even possible that one false performer (or “bad apple”) in the workplace could reduce trust between honest employees, thus creating a problematic organisational culture of low trust.

In one management discussion of the effect of False Performance behaviours on work relationships, the moderator further prompted the group, “How do you think it makes the team dynamics function?” and one manager replied by saying the following:

Destroys them. Very destructive because it breeds a hero’s trust. For the best teams to work they’ve got to trust in each other’s abilities, inabilities and be open and honest about what they’re doing, what they can do, and what they can’t do. And the moment that somebody’s acting in these manners, I think it can just erode the trust which means you’re not going to work as a team ever.
**[Management, FG05]**


A manager in another group also spoke of the negative impact on staff team morale:

I mean the cost of it all has an impact on people; the emotional cost and the monetary cost can mean that relationships get quite strained and so they can alter the whole dynamic of the workplace. You’ve only got to have one person that’s causing those sorts of problems for other people to then start losing morale.
**[Management, FG08]**


Again, these examples suggest that the false performer’s singular behaviour could affect the entire team, eradicating a climate of trust, meaning that successful teamwork then becomes impossible. One non-manager shared how the effects could also spill over into co-worker’s private lives, having themselves experienced this as a result of a false performing colleague. They explained, “…if you’re stressed, you can’t do the job...You’re angry at home all the time. You think about work, you know?” **[non-management, FG02].**

Interestingly, there was some discussion in focus groups of how the false performer can strike up strategic workplace friendships which also negatively impacts team morale as co-workers are usually more aware of the manipulative nature of this behaviour. Participants spoke about how the false performer might use work friendships to gain favour with peers or befriend superiors whom they believe can offer rewards and/or help them to advance in the organisation. When talking about the performance-appraisal review, one non-manager explained how strategic work friendships can be achieved:

…they’re [the false performer] in there with their superior, being all pally [friendly], you know. I think, again, it’s talking themselves up and befriending and keeping that rapport going with a ‘you’re my mate’ sort of attitude is what I’ve experienced in the past. These people become the best mate of their superiors and suck up to them basically.
**[Non-management, FG07]**


This description of the false performer resonates with the unfair practice of nepotism, whereby those in power show favouritism towards family or friends by, for example, appointing, promoting, or rewarding them in the workplace. In this scenario, the false performer is focused on building a friendship with their superior/s in the pursuit of a favourable performance-appraisal outcome, thus hoping that the relationship will compensate for their incompetence.

In fact, the idea of the “pally” (friendly) false performer was expanded on at length by one manager who vividly illustrated how, in their experience, the false performing “bad apple” can succeed in creating a remarkably good impression:

The individuals that I’ve encountered along the way that fit the description are normally very good communicators, very affable, nice people; people that you would like to have a chat with, and you would have a drink [with] in a bar, and all those sorts of things. Because to mask their inability or their incompetence, they mask it through their nice, pleasant demeanour, so you focus on ‘they’re a great guy, they’re a great woman, whatever.’ And that’s not based upon their competence but based upon them as a person…The alternative is, if they’ve not got that great a personality, then you tend to not focus on them as a person, so you will focus on their ability or their competence or incompetence.
**[Management, FG05]**


## 15. Preventing False Performance in the Workplace

At the end of each focus group, participants were asked to provide their solution to the problem of False Performance. They discussed appraisal software, better disciplinary procedures, and personality tests, but the main subcategories emerged as Training and 360-Degree Appraisal.

### 15.1. Training

Training was the most popular recommendation for managing False Performance. Management and non-management appeared equally interested in how training could be used to tackle False Performance. One manager suggested training managers to better monitor False Performance from the early stages:

I think the focus really should be on training managers to manage the scenario. The only reason these people exist in organisations is because managers don’t manage them, and usually…it becomes a problem. And, when you look into the individual’s past history, then everyone knows they’ve never performed but no one’s ever done anything about it. And that tends to be the problem. So, I’d want early interventions and appropriate management interaction as well.
**[Management, FG05]**


Training managers to detect and actively manage False Performance, as opposed to ignoring it, will likely help to curb False Performance before it escalates into a bigger problem for the organisation. HR managers should play a key role in facilitating such training, thus reflecting their recognition of False Performance as an issue for strategic attention and fostering a commitment to active workplace policies designed to reduce, if not eliminate, False Performance. Hailey et al. [63] also point out that much responsibility for people-focused HRM is devolved to line management, and, thus, alongside the HR department, they have a crucial role to play in guiding appropriate employee behaviour on behalf of the organisation to enhance productivity.

One non-manager strongly recommended assessment and training of all employees which should start from recruitment and remain ongoing:

Better recruitment processes, in terms of completely all the way up from doing initial assessments, training needs analysis. And then more into peoples’ ongoing developments, developing them because…a lot of this can sit in competency…whereby, people can’t do it, but they’re just going to hide behind it and say they can do this and say they can do that.
**[Non-management, FG06]**


The overall emphasis here is on establishing competency, with the suggestion that there should be more stringent selection processes, for example, including practical assessments. Given that this non-manager explained, “…a lot of this can sit in competency…”, a competency-based job interview would help to counteract the issue of candidates hiding their incompetency. This non-manager also suggested that organisations use training needs analysis (TNA) to identify performance gaps between employees’ current knowledge, skills, and abilities and where the organisation would like them to be. Beneficially, this systematic approach would help to identify and reduce False Performance in a way which would not entail singling out and stigmatising individual false performers. Finally, these recommendations underline how important it is that HR managers do not assume that all False Performance will be detected and dealt with during the recruitment process; continuous appraisal of employee performance is critical to monitoring and managing False Performance.

### 15.2. Three-Hundred-and-Sixty-Degree Appraisal

Both management and non-management participants spoke very enthusiastically about 360-degree appraisal as a means of tackling False Performance. One non-manager advocated the following:

Three-hundred-and-sixty-degree appraisal and staff surveys because then you’re getting views from different perspectives in the organisation and, if the staff surveys are anonymous, people are more likely to get some real views as to what’s going on from a different perspective, rather than just what you’re being told by the person beneath you.
**[Non-management, FG06]**


As 360-degree appraisal aids transparency by spanning the organisation to gather varied feedback on job performance from management, non-management, and other stakeholders; it is another way in which HR managers can actively seek to detect and manage False Performance. However, as highlighted by Hailey et al. [63], if HRM policies (in this case, 360-degree appraisals) are to actively impact on organisational performance, they need to be properly enacted by line management and the HR department to achieve the intended results. One manager indicated that one possible result is that a “…three-sixty appraisal system…actually forces reflection with the team” [**management, FG05**]. Whilst HR managers should avoid the enforcement of team reflection, the hope is that the wide-spanning nature of 360-degree appraisal will encourage employees to think about how they work as individuals and as a collective and, with the support of management, determine how their performance could be enhanced for the benefit of themselves, their co-workers, and the organisation.

## 16. Discussion and Conclusions

### 16.1. Theoretical Contributions

This paper represents a valuable theoretical contribution to the scarce False Performance literature, supporting previous research [1,3,4] by further establishing the construct of False Performance, as well as clarifying its relationship to job performance, interview evaluations, performance appraisal, and trust. To return to the original five research questions, the current study has successfully explored and answered these as follows:

**(1) *Furthered an understanding of the construct of False Performance and its relationship to job performance***. The results provide support for the proposal put forth in the literature review that False Performance is a new and negative addition to the taxonomies of job performance [12]. These findings are especially beneficial for organisations as they are grounded in the perceptions and experiences of real employees in the public sector, thus taking False Performance research a vital step beyond Parnell and Singer’s [1] initial self-generation of OCS items. Notably, the current study found some evidence for items in the OCS; for instance, the emphasis placed on dressing to impress (which, ultimately, did not emerge as a grounded-theory category) and a preference for looking busy rather than being busy. However, the plethora of categories and sub-categories reported suggest that there is a much wider range of False Performance behaviours than initially identified by Parnell and Singer [1], demonstrating the value of the focus-group methodology and grounded-theory approach for generating new False Performance behaviours.

Also, the current study provides further evidence for the relationship between IM techniques and job performance. It has moved beyond the simple cataloguing of IM techniques and focused on the critical issue of performance, as highlighted by Parnell and Singer [1]. For example, the results contain many examples of how the false performer behaves to create an impression of themselves as competent, whilst actually performing incompetently on the job. This juxtaposition is highlighted well by the category of Co-Worker Perceptions of False Performance, whereby employees ably discussed how false performers engage in various forms of subterfuge to impress management whilst usually doing so at the expense of their co-workers (who often have to pick up the slack). Thus, the current paper consolidates Parnell and Singer’s [1] description of False Performance as being related to IM but still conceptually distinct and supports the False Performance literature [4], which suggests that False Performance is probably more recognisable by co-workers.

**(2) *Clarified how False Performance operates in the job interview and the performance-appraisal review.*** By using grounded-theory and focus-group methodology, participants spoke freely and candidly about False Performance and the role it plays in interview and performance-appraisal settings. Just as IM has been shown to have a significant effect on interview evaluations [17] and performance-appraisal ratings [18], the current results indicate that False Performance has a similar effect upon interviewer perceptions. It appears that the false performer’s behaviour helps them to elicit favourable evaluations from the interviewer, despite their underlying incompetent job performance. Given the potential impact of job interviews and performance-appraisal reviews on the success of HRM, it is recommended that HR managers and management take note of these findings and ensure that they safeguard against False Performance in formal settings.

In the literature review, one question posed was, “Can False Performance ever be positive?” Whilst the results indicate that False Performance can result in some positive outcomes for the false performer (e.g., success at interview and being promoted beyond their competence levels), the qualitative data strongly reinforce the definition of False Performance as a negative workplace behaviour, designed only to give the illusion of competent job performance. Focus-group discussions offer remarkably open insights which demonstrate that there are unlikely to be any positive outcomes of False Performance for the false performer’s co-workers or the organisation. The results also strongly support the argument that False Performance is not regarded as a legitimate political skill, i.e., one which incorporates positive and honest work behaviours [30].

**(3) *Explored the relationship between trust in the workplace and False Performance behaviours***. Based on previous False Performance research [3], the current study sought to explore whether trust negatively or positively affects False Performance. Certainly, some of the qualitative findings support the recent research [38], which contends that the higher the trust an employee has in management, the better their job performance is likely to be. Similarly, our findings also offer tentative support for Gbadamosi et al.’s [4] research, which found that the lower the level of an employee’s trust in management, the lower their likely commitment and the greater the likelihood they will, therefore, engage in False Performance. The current paper has, therefore, made significant strides towards bridging the lengthy gap since the original False Performance research [1] and reexploring its relationship to trust [3,4] with reference to more recent, related research [38]. However, the picture related to this research question emerged as more complex than anticipated, as is evident from the main category of The Impact of Trust on the False Performer.

Based on the current results, no definite conclusions can be drawn, as the qualitative data simultaneously suggest the following: (a) co-worker trust will have no effect on False Performance (i.e., subcategory of Trust is not a concept for the false performer), reminiscent of Gbadamosi’s [3] finding that there is no significant relationship between false performance and trust; (b) co-worker trust will negatively affect False Performance (i.e., the false performer will exploit trusting co-workers and false perform more), consistent with Yakovleva et al.’s [48] explanation that low trusting individuals will frequently become more exploitative of longstanding, cooperative/trustworthy co-workers); and (c) co-worker trust will positively influence False Performance (i.e., the false performer will feel shame/guilt about false performing and, therefore, false perform less).

Whilst none of the focus groups specifically identified False Performance as a predictor of shame proneness [50], the findings indicate that feelings of shame and/or guilt could operate as possible moderators of the relationship between co-worker trust and False Performance. Overall, given the present study’s conflicting results around trust and False Performance, we strongly recommend that future studies clearly articulate the research question around this relationship, for example, by examining the relationship between trust in management and False Performance versus the relationship between co-worker trust and False Performance.

**(4) *Examined the impact of False Performance on co-worker relationships***. Unsurprisingly, focus-group participants spoke about the demoralising effects of False Performance on the rest of the team, with some questioning why they should put effort into their job performance when the false performer gets away with doing less but still attracts rewards. There was the worrying suggestion that the false performer’s behaviour could lead to a contagion effect, whereby co-workers will be tempted to engage in False Performance to receive similar rewards. Co-workers also spoke about the impact of False Performance on their personal wellbeing, both in the workplace and at home (where it can be hard to switch off from work stress). And there was consideration of how co-worker trust in management could be damaged by the failure of the organisation to tackle False Performance.

The results also encompass some behaviours which could be interpreted as social loafing (e.g., exercising less effort in a group). However, to reiterate the findings of the literature review [34], social loafing is an unconscious and naturally occurring phenomenon, whereas the False Performance behaviours discussed in focus groups refer to conscious behaviours designed to deliberately mask incompetence and create the illusion of competence. In the case of the false performer, they deliberately withhold effort because they know that the team’s competent job performance will help to compensate for their incompetence. Future studies of False Performance should maintain an emphasis on the conscious nature of False Performance, preferably with reference to Table 1, in which individuals who are high in self-presentation but low in competence are defined as false performers.

We also recommend that any future studies focused on the impact of False Performance on co-worker relationships examine the role of social media [39]. Kasim et al.’s [39] recent research suggests that social capital (online social network ties and trust) positively promotes work engagement and job performance. The current study did not yield any significant data on the false performer’s online work relationships; however, this may be because the current study was conducted before the COVID-19 pandemic, when online connections in the workplace were not so commonplace. Considering the huge increase in remote working (where employees work from a remote location outside the office/workplace) and expansion of social network ties, this would be a fascinating area to explore. Based on Kasim et al.’s [39] research, there could be an exploration of whether the false performer’s social media usage significantly predicts online social network ties and co-worker trust and if this is then able to positively impact on their job performance.

**(5) *Generated and co-created practical solutions to prevent/reduce False Performance in organisations.*** Through the category of Preventing False Performance in the Workplace, focus-group participants generated potential solutions for the prevention and reduction of False Performance. For example, training and 360-degree appraisal were both discussed, with a specific focus on training managers to manage the scenario and a suggestion that management should actively intervene when they recognise that an employee is false performing, rather than ignoring this type of negative behaviour. To refer back to the literature review, Ergün’s [35] research findings highlight how mushroom-type management behaviour might lead to teachers experiencing loneliness and exhibiting more False Performance behaviours in the school environment. Whilst the results of the current study do not reference mushroom-type management, they do indicate that a superficial management style which leaves employees in the dark (e.g., a manager who ignores False Performance and avoids discussing it with the team) may well promote a culture which allows False Performance to thrive. Therefore, a transparent approach to managing and communicating sensitively about False Performance in the workplace could well help to identify and eliminate/reduce it.

### 16.2. Managerial Implications

Following on from the theoretical findings, this paper encourages organisations and managers to consider how they can implement HRM initiatives to better detect and manage False Performance amongst their workforce. Gbadamosi [3] has previously warned that, if negative False Performance behaviour is ignored, there may be a decline in morale, as employees increasingly feel a sense of unfairness. This is supported by the current study, which found evidence that False Performance can affect the productivity and wellbeing of other employees. To refer back to the five research questions again, managerial implications which have emerged from the results include the following:


**
*(1) An improved managerial understanding of the construct of False Performance and its relationship to job performance.*
**


Throughout this paper, there has been a continuous effort to enhance HR manager’s understanding of False Performance. The first step to reducing and eliminating False Performance is for managers to clearly understand what the construct is and how it negatively affects job performance. We have provided a comprehensive literature review which extends well beyond Parnell and Singer’s [1] original research to provide a more comprehensive understanding of the field [6,35,50]. We urge HR managers to familiarise themselves with both the historical and up-to-date context of False erformance in a pre-emptive effort to recognise and reduce False Performance in the workplace.


**(2) *Increased ability to identify False Performance in the job interview and the performance-appraisal review.***


Critically, the current study alerts organisations and managers to the need to examine their own recruitment and selection processes, performance-appraisal systems, and organisational culture. Our proposed introduction of False Performance into HRM policies has a number of vital practical implications, namely improved recruitment processes (e.g., competency-based interviews) and fairer one-to-one performance-appraisal reviews. By equipping managers with a better understanding of False Performance, combined with an increased insight and ability to design better recruitment and selection processes, this should help to prevent the false performer from gaining access to the organisation. The category of Preventing False Performance in the Workplace offers additional recommendations for how HR managers can identify and tackle False Performance amongst existing employees, but this is by no means an exhaustive list.

**(3) *An insight into the potential relationship between trust in the workplace and False Performance behaviours***.

There has been extensive consideration of the relationship between trust and False Performance in the literature review and discussion [3,4]. Whilst the relationship between the two variables is complex, as evidenced by the current study’s focus-group results, the emerging picture provides compelling evidence to suggest that HR managers should aim to reduce/eliminate False Performance to maximise harmonious and trusting relationships amongst co-workers. Moreover, the evidence suggests that authentic management, characterised by trust in the supervisor/organisation, is likely result in better job performance [38]. Therefore, based on the recent literature and current paper, we implore management to prioritise fairness in their managerial methods, actions and activities so that all employees are encouraged to perform as best they can (admittedly, with a question mark over whether the false performer’s behaviour will be positively influenced by trust in management).

Furthermore, based on recent research into the impact of remote working on job performance [44], we further advise managers to invest in their online communication and management skills. Managing employees remotely is very different to managing employees in a physical workplace and management needs to be aware of the challenges inherent in remote working. For example, it is more difficult to physically see and, therefore, monitor employees’ work behaviour, and it is often harder to develop trusting relationships with subordinates who are rarely seen in person. Kifor et al.’s [44] findings suggest that employees with low trust in management are more likely to rely on co-workers to get the job done. The current paper offers a wealth of real-life examples to demonstrate how the false performer is already prone to exploitation of their co-workers. Therefore, managers should strive to engender employee trust to safeguard against, inadvertently, encouraging false performers to rely even more heavily on their co-workers.

**(4) *An appreciation of the negative impact of False Performance on co-worker relationships***.

The subcategory of Reluctance to Report False Performance illustrates how a manager who overlooks False Performance or penalises employees for reporting co-worker’s False Performance is likely to create bad team morale. In the focus groups, employees discussed their frustrations with management who avoid managing false performers. We encourage managers to take note of this finding so that they avoid becoming like the mushroom-type managers, who keep their employees in the dark [35], for example, by ignoring false-performing subordinates, failing to constructively tackle their negative behaviours, and keeping other employees in the dark (like mushrooms) by refusing to discuss incidents of False Performance. In line with one of the main solutions to emerge under the category of Preventing False Performance in the Workplace, if managers are properly trained to identify, manage, and understand the fallout from False Performance, this will help to create a healthier workplace in which authentic and competent job performance is shown to be valued; and, consequently, trust in management, co-worker trust, and organisational culture are all likely to improve.

Furthermore, transparent management of False Performance is likely to mitigate against the contagion effect, which was discussed as part of the category of The Effect of False Performance on Co-Worker Morale. One focus-group participant explained, “If a false performer seems to be doing well, it can make you want to false perform.” This finding may be contextualised by reference to Felps et al. [64] who explain that “bad apples spoil the barrel” (p. 175), whereby one individual’s dysfunctional behaviour can impair the functioning of the entire group. Thus, initially, functional group members may start to withhold effort to restore equity with the “bad apple” [64]. Consequently, this paper urges managers to actively address the problem of False Performance to ensure that this negative behaviour does not spread amongst the workforce.


**(5) *A pro-active approach to facilitating practical solutions to prevent/reduce False Performance in the workplace.***


Finally, and perhaps foremost, this paper encourages HR managers to take a pro-active approach to facilitating practical solutions for the prevention and reduction of False Performance in the workplace. The category of Preventing False Performance in the Workplace offers two initial suggestions for preventing and reducing False Performance, i.e., training and 360-degree appraisal. However, these are just starting points from which to tackle the phenomenon of False Performance, and, as such, we urge HR managers to be innovative in facilitating solutions to False Performance and, indeed, to take inspiration from the focus-group and grounded-theory methods employed in the current study so that they firmly “ground” their solutions to False Performance in feedback collated from employees.

However, HR managers should also take heed of Lewandowski’s [6] warning that co-production could be co-contaminated by false performers and, therefore, proceed with caution if they do take a co-production approach towards tackling False Performance in the workplace. They should remain vigilant for false performing employees deliberately inputting information which might co-contaminate the process. Co-contamination is anything harmful added to that which is wholesome that could then render a service, process, or product unpalatable in some way [7]. This could lead to “co-destruction” of value through the interactions between different service systems [65]. The main implication of co-destruction for management is how the disclosure of negative performance, such as False Performance to the public, could impact people’s trust in the organisation. As such, managers should take care to prevent and reduce False Performance to maintain the trusted reputation of their organisation.

### 16.3. Limitations and Future Directions

The current study has certain limitations, one being the results, which cannot be generalised to the private or voluntary sector. The research was conducted in the public sector based on the literature indicating a greater incidence of False Performance in the public sector, where lower levels of trust have been found. The fact that the current study found trust to be operating in various directions and opposing ways in relation to False Performance suggests the need for an alternative rationale for sample selection. Nevertheless, based on the conjectured differences between sectors, it is recommended that a future comparative study examine False Performance in the public versus the private sector to enhance the contextual understanding of False Performance. For example, the average age of the sample in each sector may be quite different, a variable which may affect the results. In the current study, the average age of participants (48 for management; 40 for non-management) reflects the median age of 44 years old for UK civil servants [66]. However, this may not be characteristic of the private sector, where jobs are skewed towards the younger age groups [67]. In the current study, the benefit of mature average age and lengthy tenure (29.7 for management; 22.7 for non-management) was that it likely contributed more work-situated experiences to the data.

The results indicate that False Performance may differ depending on work situation, suggesting that there may be different dimensions of False Performance related to organisational context. The present study illustrated that, in the job interview, the false performer is a newcomer to the organisation and, as such, has more opportunities to false perform. In contrast, in the performance-appraisal review, the false performer is an existing employee whose behaviour has been observed for some time, so they must, necessarily, employ different False Performance tactics. There is also another dimension to consider, that of False Performance in the everyday workplace, where co-workers are better positioned to observe the false performer’s everyday job performance. It would be useful for future studies to examine how False Performance in the high-stakes context of the job interview or performance-appraisal review differs from False Performance on the job. Such research would help to clarify whether False Performance is a multidimensional construct.

In the Introduction, the negative construct of False Performance, which masks incompetence, was distinguished from the positive construct of political skill, which facilitates promotion. The qualitative data suggest that False Performance also has a natural association with promotion in the organisation, with many examples of the false performer using deceptive tactics to rise through the ranks. Thus, False Performance seems to serve two aims: to get ahead in the organisation (although speculative, this may be driven by a desire for power or a better salary) and/or to mask incompetence (again, whilst speculative, this may be driven by the false performer’s desire to retain their job). As the construct of False Performance evolves, there is a need for future studies to distinguish between the drivers behind these two aims, with a recognition that they could, of course, co-occur.

Future research could also investigate other potential drivers of False Performance. Previous research [3,4] has identified several variables which may correlate with False Performance (e.g., employee commitment, core self-evaluation, and turnover intention). Furthermore, there are yet unexamined variables which could be hypothesised to predict False Performance (e.g., personality, organisational hierarchy, and leadership style). In future qualitative studies, the skilful use of open-ended questions could help to bring new ideas to the fore about these and additional variables. Future quantitative investigation could also be used for this purpose, as well as the exploration of potential moderators in the relationship between trust and False Performance. Some of these were discussed in the subcategory of Co-Worker Trust Reduces False Performance, but others may include the following: (1) the private versus public sector—looking at whether the relationship between trust in management (or trust in the organisation) and False Performance could be moderated by sector type to affect levels of False Performance; and (2) evaluation versus everyday context—looking at whether the relationship could be moderated by evaluation context (e.g., the job interview) versus the everyday context (e.g., the office).

In terms of the global context, although the current research was conducted in the UK, there is evidence to suggest that these findings could extend to international HRM. Although there has been very little research on the construct of False Performance globally, the extant data that have been reported have come from both Western [1] and non-Western contexts [3,4], thus suggesting that False Performance is ripe for exploration on an international scale. Such research may also reveal whether culture plays a role in False Performance, helping to provide a valuable cross-cultural understanding of this phenomenon to aid effective implementation of research outcomes within organisations.

As suggested in the Introduction, the current results indicate that there is potential for expansion of the OCS. By generating False Performance data fully grounded in participants’ own perceptions and experiences, it was found that, although participants briefly discussed some issues which appear in the OCS (e.g., dressing to impress), they introduced a lot of False Performance behaviours not in the OCS (e.g., boss over-delegation to subordinates). Therefore, it is recommended that future research use the current qualitative results to revise the quantitative OCS, creating a more robust and reliable measurement instrument. Moreover, it is recommended that future investigations take account of the fact that the OCS may not be a unidimensional scale. Just as job performance has been shown to have various taxonomies [12], the current study indicates that False Performance could, likewise, consist of different dimensions.

Finally, the results reported in this paper reflect False Performance in a pre-pandemic workplace. However, the post-COVID working landscape has much more scope for a wider range of False Performance behaviours in light of the increased remote and flexible working arrangements which became commonplace during the pandemic. It is not hard to imagine how remote working could give the false performer a myriad of new ways in which to engage in negative work behaviours (e.g., using the increased technology to support their illusion of competence by being present on screen whilst not doing much off screen). This problem is compounded by the organisation’s lack of ability to observe and tackle False Performance in the same way as would be possible if the false performer were physically present at work. When working from home, the false performer is far more likely to be able to evade detection by co-workers, meaning that co-workers are no longer a reliable means of identifying false performers. Therefore, there is an onus on future research to explore how False Performance has evolved since COVID-19 and how HR managers might be better equipped to identify and tackle it in its new forms.

## Figures and Tables

**Table 1 behavsci-14-00985-t001:** The Impression Management–False Performance model of self-presentation behaviours.

High self-presentation	**False performers**	**Impression managers**
	*Undesirable employee*	*Undesirable/desirable employee* (dependent on the organisational context)
Low self-presentation	Transparent self-presentation	Transparent self-presentation
	*Undesirable employee*	*Desirable employee*
	Low competence	High competence

**Table 2 behavsci-14-00985-t002:** Demographic information according to management and non-management status.

Focus Group	Total No.	Female	Male	Mean Age (Years)	Mean Years in Organisation
Management	26	13	13	48.1	29.7
Non-management	25	17	8	40	22.7

**Table 3 behavsci-14-00985-t003:** Major categories, subcategories, and summary of focus-group narratives on False Performance.

Major Categories and Subcategories	Summary of Focus-Group Narratives
** *Co-Worker Perceptions of False Performance in the Workplace* **	
Claiming Credit for Others’ Work	Taking credit for work done by co-workers
Boss Over-Delegation to Subordinates	Boss over-delegating work to subordinates to mask their own incompetence
Shifting the Blame	Blaming other people/factors for mistakes
** *False Performance in Interview/Appraisal Settings* **	
Lying About Qualifications	Exaggerating or lying about qualifications
Over-Talking as a Smoke Screen	Over-talking to prevent the detection of False Performance
Claiming Credit for Others’ Work (1:1)	Claiming credit for others’ work in the job interview/performance-appraisal review
** *The Impact of Trust on the False Performer* **	
Trust is Not a Concept for the False Performance	The false performer will not be affected by trust
Co-Worker Trust Breeds False Performance	Co-worker trust leads to an increase in False Performance
Clarifying how Trust Relates to False Performance	Unsure of relationship between trust and False Performance
Co-Worker Trust Reduces False Performance	Co-worker trust leads to a decrease in False Performance
** *The Effect of False Performance on Co-Worker Morale* **	
Reluctance to Report False Performance	Co-workers fear reporting False Performance to a manager
Bad for Morale	Competent co-workers feel demoralised
** *Preventing False Performance in the Workplace* **	
Training	Training in detection and management of False Performance
360-Degree Appraisal	Gaining multiple stakeholder views via 360-degree feedback

## Data Availability

Data are contained within the article.

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
