# Peer review of "The Illusion of Competence: A Qualitative Deep Dive into Workplace False Performance"

_behavsci, 2024, doi:10.3390/bs14110985_

Round 1

Reviewer 1 Report

Comments and Suggestions for Authors

I would like to thank for the opportunity to review this manuscript. The topic of workplace false performance is interesting and novel. Since not much research has been done in the field, the reviewed paper can be considered a contribution to current knowledge in the field.

The paper is well-written and well-structured, as for its content no significant changes are needed. I have summarised my thoughts and recommendations.

Introduction: This section is well-written and highlights the importance of the investigated problem. The authors identified gaps in the current knowledge and clarified how the study aims to fill these gaps.

-        I recommend including all five research questions.

Literature review: A review of recent scholarship is provided, and previous research is analysed.

-        Although considering the fact that not many relevant studies have been published, I suggest to include some more recent resources (2020-2024).

Method: In the Method section, sufficient information about the research design, sample, and procedures is provided.

Results and Interpretations: The obtained results are thoroughly described and analysed.

Discussion and Conclusion

-        I recommend comparing the present findings with more recently (2020-2024) conducted international studies (if available).

I also suggest to check the applied citation and referencing style.

Reviewer 2 Report

Comments and Suggestions for Authors

The Illusion of Competence: A Deep Dive into Workplace 2 False Performance

Dear authors,

The article has many strengths, e.g. it is well written, the topic is innovative and has strong  theoretical implications. In addition, qualitative studies generate value. Finally, the authors seek to go into depth, i.e. they make a significant effort to explain every detail. However, they include a very high amount of relationships and information that may confuse the reader. This is just a reflection that does not detract from the great work done by the authors. Overall, it is a brilliant and carefully crafted piece of work.

Abstract

The abstract is well drafted and gives a good overview of the study.

1.      Introduction

Perceived performance that is far from actual performance may indicate lack of acceptance and personality disturbances. I like the term false performance better. It is true that a false performer can have a negative impact on the work climate and the well-being of co-workers. An incompetent person is promoted in an incompetent company.

It is very interesting and necessary to deepen the understanding of false performance as a construct.

The fact that this is the first research on false performance in the UK is a great novelty for the existing knowledge on the subject. I think it is important that you explain more about focus groups or at least define the term. Many potential readers will not know what a focus group is.

For your peace of mind, in any country in the world, false performance is most prevalent in public companies. It is a paradox since public companies should hire people with a high perception of social responsibility. The relationship between low trust and false performance is very interesting. Building an environment of trust is essential for any organisation, public or private. In any case, we could establish a relationship between abuse of trust and false performance.

The different Cronbach's alphas between the original authors and some later authors are strange. It is good that they point out these differences as it is possible that the scale does not have the necessary reliability.

They relate Parnell and Singer's study to their own in a good way. In addition, including qualitative data is always a good thing.

The problem and the general objective are well defined. Objective: to improve understanding of the construct of False Performance, as well as to explore the relationship between False Performance behaviour and job performance. The five research questions are clear. I would believe that they even go beyond the general objective.

It would be interesting to know what your research brings in relation to previous research regardless of the country.

2.      Literature Review

2.1. Human Resource Management

It is true that understanding and addressing false performance is crucial for Human Resources (HR) professionals. This part is well written and correct.

2.2. Job Performance

Proposing false performance as a new taxonomy of job performance is interesting. The difference between false performance and counterproductive performance is correct. However, false performance does not only hide incompetence. False performance goes far beyond incompetence itself. This part is well written and correct.

2.3. False Performance versus Impression Management

The differences between MI and false performance are interesting. The possible links between the two variables are also interesting. The common denominator so far is the use of very old bibliographic references, why?

3.      False Performance and Mushroom-Type Management

The relationship between mushroom-type managers, loneliness and false performance is very interesting. However, the relationship between superficiality, false performance and climbing the hierarchical ladder is hard to believe because of the low interaction.

4.      Unethical Work Behaviour Literature

5.      The False Performer as a ‘Good Actor’

It is difficult to interpret the selfless actions of good soldiers against the selfish intentions of good actors. Organisational behaviour is generally selfish; therefore, it clashes with the emotional intentions of good soldiers.

6.      Can False Performance Ever Be Positive?

It is very important to differentiate political skills from false performance.

7.      Social Loafing

It is important to differentiate people who hold back their competences from people who lack competences.

8.      Definitional Issues

Table 1. The Impression Management-False Performance Model of Self-Presentation Behaviours (It is well prepared and well understood).

Continuance commitment has very different results when analysed unidimensionally or across its two dimensions (lack of job alternatives and perceived costs of job separation). Undoubtedly, unidimensional continuance commitment is positively associated with false performance. Very interesting is the inverse relationship between trust and false performance. Re-examining the relationship between trust and false performance is very important. Finally, the relationship between shame and false performance is interesting.

9.      Method

The choice of a qualitative study through focus groups is a wise one. Moreover, it is unusual in terms of the effort required.

9.1.  Participants

There is no strong bias in the sample. Moreover, the average length of service is very high. Table 2 is fine

9.2.  Data Collection

The process is correct. Limiting the number of participants in the focus groups is a good thing. Looking for similarities and comfort among focus group participants is another good thing. Overall, very good job.

9.3.  Data Analysis

I was unaware of the grounded theory. The theory is explained in detail. The process is correct as is the qualitative data analysis software used.

10.   Results and Interpretations

In the public sector, incompetence is often rewarded. In other words, relationships are valued more than skills (nepotism). Selection processes should be stricter and include practical tests and practical group tests. In addition, the interview with the candidate should be competency-based. The different categories used are fine with me. Table 3 is clear and correct.

11.   Co-Worker Perceptions of False Performance in the Workplace

11.1.                 Claiming Credit for Others’ Work

11.2.                 Boss Over-Delegation to Subordinates

11.3.                 Shifting the Blame

The first block of results: ‘Coworkers’ perceptions of false performance in the workplace’ is explained in detail. In addition, the inclusion of anecdotes and stories is one of the advantages of a qualitative and focus group study.

12.   False Performance in Interview/Appraisal Settings

12.1.                 Lying About Qualifications

12.2.                 Over-Talking as a Smoke Screen

12.3.                 Claiming Credit for Others’ Work

13.   The Impact of Trust on the False Performer

13.1. Trust is Not a Concept for the False Performer

13.2. Co-Worker Trust Breeds False Performance

13.3. Clarifying How Trust Relates to False Performance

13.4. Co-Worker Trust Reduces False Performance

The second and third blocks are also very well explained. Indeed, qualitative studies need this level of detail. I have no suggestions for this whole part as the authors are thorough and have a good level of writing.

14.   The Effect of False Performance on Co-Worker Morale

14.1.                 Reluctance to Report False Performance

14.2.                 Bad for Morale

15.   Preventing False Performance in the Workplace

15.1.                 Training

15.2.                 360-Degree Appraisal

Exactly the same with the fourth and fifth blocks.

16.   Discussion and Conclusión

16.1. Theoretical Contributions

Regarding the theoretical contribution, it would be interesting to include much more recent publications, whether qualitative, quantitative or mixed. They cite authors who are nearly 20 years old. It is true that this research goes beyond that elaborated by Parnell and Singer. I would have liked a discussion in which they debated their results with respect to the results of other authors. Theoretical contributions only describe their results without knowing exactly what knowledge gap they have managed to fill. Fundamentally, compared to much more recent studies. They cite some works to establish certain comparisons or to explain their findings, but the discussion could have a greater impact considering the degree of complexity of their research. They could enrich the answer to the five research questions. Their work is very good. I'm just trying to make it a little better.

16.2. Managerial Implications

The practical implications could be improved by following the pattern used in the theoretical implications. That is, each research question could have its practical implications and even the general objective could have its own practical implications. I think this is one of the most important parts of research, the practical implications.

16.3. Limitations and Future Directions

In general, they specify limitations and future research well.

Round 2

Reviewer 2 Report

Comments and Suggestions for Authors

Dear authors,

Thank you very much for this new version of your article, after reading it carefully I was pleasantly surprised.

Best regards.

Dear Editor,

As I told you a few weeks ago, the article has many strong points. The topic is innovative, well written and has strong practical implications. In addition, the authors are very detail oriented and make a profound effort to ensure that the reader is at all times contextualized.

The authors improve the introduction. They better define focus groups and explain in greater detail the rationale for using this method. In addition, the authors incorporate a significant number of new references that go beyond Parnell and Singer (2001). The authors establish a clear differentiation between false performance and political skills. The authors introduce continuity commitment in greater depth.

The managerial implications are reworked largely in response to the various suggestions. The discussion and conclusions are also improved to enhance the five research questions in the study. Overall, it is a good article, and the authors have made a great effort to improve it.

If you have any questions, please do not hesitate to write to me

Best regard